# FGFR2 Mutation p.Cys342Arg Enhances Mitochondrial Metabolism-Mediated Osteogenesis via FGF/FGFR-AMPK-Erk1/2 Axis in Crouzon Syndrome

**DOI:** 10.3390/cells11193129

**Published:** 2022-10-05

**Authors:** Yidi Wang, Yue Liu, Haotian Chen, Xiaojing Liu, Yi Zhang, Yixiang Wang, Yan Gu

**Affiliations:** 1Department of Orthodontics, National Engineering Laboratory for Digital and Material Technology of Stomatology, Beijing Key Laboratory of Digital Stomatology, Peking University School and Hospital of Stomatology, Haidian District, Beijing 100081, China; 2Department of Oral and Maxillofacial Surgery, Peking University International Hospital, Changping District, Beijing 102206, China; 3Department of Oral and Maxillofacial Surgery, National Engineering Laboratory for Digital and Material Technology of Stomatology, Beijing Key Laboratory of Digital Stomatology, Peking University School and Hospital of Stomatology, Haidian District, Beijing 100081, China; 4Central Laboratory, Peking University School and Hospital of Stomatology, Haidian District, Beijing 100081, China

**Keywords:** craniosynostosis, constitutive activation mutation, osteogenic differentiation, mitochondrial metabolism, AMPK

## Abstract

Background: Crouzon syndrome ([OMIM] #123500) caused by FGFR2 mutation is an autosomal dominant syndrome with craniosynostosis, the underlying mechanism of which remains obscure. Methods: First, whole exome sequencing was used to screen the possible pathogenic variant in two sporadic patients with Crouzon syndrome. The investigation of primary and secondary structures as well as the conservation analysis of FGFR2 mutation (p.Cys342Arg) was performed. Then, wild-type and mutant overexpression plasmids were constructed and transfected into pre-osteoblastic murine cell line MC3T3-E1 cells. Osteogenesis and mitochondrial metabolism were analyzed by CCK8, ALP staining and ALP activity, alizarin red staining, qRT-PCR, Western blot, seahorse assays and mitochondrial staining. The siRNA targeting FGFR2 and domain negative FGFR2 were designed for verification. Results: First, FGFR2 mutation (p.Cys342Arg) was detected in two sporadic Chinese Crouzon syndrome patients. FGFR2 p.Cys342Arg promoted the osteogenic differentiation of MC3T3-E1 cells through the upregulation of AMP-activated protein kinase (AMPK)-Erk1/2 signal pathway. Furthermore, FGFR2 p.Cys342Arg enhanced oxidative phosphorylation and converted mitochondrial fusion to the fission of MC3T3-E1, promoting osteogenic differentiation and craniosynostosis in Crouzon syndrome. Additionally, AMPK or Erk1/2 inhibitors delayed the cranial suture closure. Conclusion: FGFR2 mutation p.Cys342Arg promotes osteogenesis by enhancing mitochondrial metabolism-mediated via FGF/FGFR-AMPK-Erk1/2 axis, which indicates the potential of therapy targeting AMPK or Erk1/2 for syndromic craniosynostosis treatment.

## 1. Introduction

Crouzon syndrome (Online Mendelian Inheritance in Man (OMIM) #123500) is an autosomal dominant disease with an incidence of 1 in 62,500 characterized by craniosynostosis, midfacial hypoplasia with crossbite, shallow and large orbit, eagle nose and other accompanying symptoms [1,2]. Fibroblast growth factor receptor 2 (FGFR2) is considered as the genetic etiology of Crouzon syndrome [3,4,5,6,7]. In addition to FGFR2, FGFR3 mutation is thought to be related to Crouzon syndrome with acanthosis nigricans ((OMIM) #612247) [8,9]. 

FGFR is a kind of highly conserved transmembrane receptor composed of three extracellular immunoglobulin-like domains, a transmembrane domain and a cytoplasmic tyrosine kinase domain that acts as a tyrosine-protein kinase and combines with at least 18 kinds of FGF ligands along with varying degrees of sequence similarity [10,11,12]. The FGF/FGFR signaling pathway plays an essential role in the regulation of cell proliferation, differentiation, migration, and especially the osteogenesis process and embryonic development. Classic downstream signals of the FGF/FGFR signaling pathway include: Erk1/2 MAPK signaling pathway, p38 signaling pathway, JNK signaling pathway, phospholipase C γ (PLCγ) signaling pathway, and protein kinase C (PKC) signaling pathway [13,14,15,16]. 

Most FGFR2 mutations are located in the Ig-III domain that has been proven to be a genetic contributor to Crouzon syndrome through the activation of one or more FGF/FGFR downstream signaling pathway(s) [17,18,19,20]. However, to date, there is scarcely any effective etiological treatment for Crouzon syndrome, and therapeutic methods are confined only to surgical treatment such as cranial vault decompression and reconstruction, midfacial advancement, and symptomatic treatment for a series of accompanying symptoms [21]. It is urgent to that the underlying mechanism of FGFR mutation-caused Crouzon syndrome is extensively investigated to develop a new way of intervening in the disease process. 

In this study, we found the new underlying mechanism whereby FGFR2 mutation p.Cys342Arg enhances mitochondrial metabolism-mediated osteogenesis through the FGF/FGFR-AMPK-Erk1/2 axis in Crouzon syndrome, which may provide a new adjuvant therapy for treating Crouzon syndrome.

## 2. Materials and Methods

### 2.1. Participants and Clinical Evaluations

Two sporadic patients with Crouzon syndrome were diagnosed by surgeon specialists and stomatologists from Peking University School and Hospital of Stomatology and Peking University International Hospital. Both patients underwent sequential treatment at Peking University International Hospital. Blood samples, cone beam CT and digital photos were taken from the two patients with informed consent.

Another 50 healthy controls were free of Crouzon syndrome and other bone or cartilage hereditary diseases. Blood samples or pharyngeal swab samples were taken from the 50 controls. Our work was approved by the Biomedical Ethics Committee of Peking University School and Hospital of Stomatology (PKUSSIRB-202056089).

### 2.2. Mutation Screening and Pathogenic Analysis

Whole-exome sequencing (WES) was conducted to detect the genetic mutation of the two patients and Sanger sequencing was utilized to verify the authenticity and accuracy of mutation. Quality control, alignment analysis and the calling analysis of WES data were shown in Appendix A. Primary and secondary structures as well as conservation analysis were conducted to predict the possible impact of mutations on the structure and function of FGFR2. More details of the process are shown in the Appendix A.

### 2.3. Cell Culture, Transfection and Osteogenic Induction

Pre-osteoblastic murine cell line MC3T3-E1 cells were cultured under aseptic conditions using high-glucose Dulbecco’s modified Eagle medium (DMEM, Hyclone, Logan, GA, USA), containing 10% fetal bovine serum (FBS, Biological Industries, Israel) and 50 U/mL penicillin and 50 μg/mL streptomycin (Hyclone, Logan, GA, USA) in a humidified atmosphere of 5% CO_2_ at 37 °C. Cells were seeded at 6~8 × 10^4^ cells/well in 12-well plates. 

To knock down FGFR2, small interfering RNA (siRNA) specifically targeting FGFR2 were designed, (sense, GGAGUUUAAGCAGGAGCAUTT, antisense, AUGCUCCUGCUUAAACUCCTT) and siNC was used as a negative control (Shanghai Lingke Tech, Shanghai, China), transfected into MC3T3-E1 cells using JetPRIME transfection reagent (Ployplus-transfection, Illkirch, France), and incubated for 48 h. At 48 h post-transfection, the medium was replaced by osteogenic differentiation induction medium (Cyagen, Suzhou, Jiangsu, China).

As shown in Appendix A, wild-type (WT) FGFR2, Cys342Arg mutant (MT) FGFR2 and domain negative (DN) FGFR2 overexpression plasmids with Flag tag on the side of C-terminal were constructed by Tsingke Biotechnology Co, Ltd, China. Additionally, they were extracted by EndoFree Plasmid Midi Kit (Jiangsu Cowin Biotech Co, Jiangsu, China) and then transfected into MC3T3-E1 cells using JetPRIME transfection reagent and incubated for 24 h.

### 2.4. Cell Proliferation Assay

The Cell Counting Kit-8 assay was used to detect cell proliferation. The transfected cells were seeded into 24-well plates at a density of 3000 cells/well. Cell viability was tested by the CCK-8 system (Dojindo, Kyushu, Japan) at 6, 12, 24, 36, 48, 56 and 60 h after cell inoculation according to the manufacturer’s instruction. Each test with three or more duplicated wells was repeated at least three times.

### 2.5. Quantitative Real-Time PCR

The total RNA of cells was extracted using TRIzol reagent according to the manufacturer’s instruction (Invitrogen, Carlsbad, CA, USA). Primers were designed using Primer3 and BLAST to amplify 80-220 bp within the coding sequences of target genes and ribosomal protein S18 (*RPS18*, housekeeping control). After 5 days of in vitro osteogenic induction, the expression of osteogenic genes, collagen type Ⅰ α 2 (*ColIα2*), RUNX family transcription factor 2 (*Runx2*), alkaline phosphatase (*Alp*), osteocalcin (*Ocn*) and osteopontin (*Opn*), mitochondria dynamic-related genes mitofusin 2 (*Mfn2*), OPA1 mitochondrial dynamin-like GTPase (*Opa1*) and dynamin-related protein 1 (*Drp1*) were detected at the transcription level. The sequences of the primers and the expected size of the PCR products are shown in Appendix A. All PCR reactions were performed in a 7500 Real-Time PCR machine using Taq Pro Universal SYBR Green qPCR Master Mix (Vazyme, Nanjing, Jiangsu, China). The PCR parameters for the target genes were as follows: 40 cycles for denaturation at 95 °C for 5 s, extension at 60 °C for 1 min. The expression levels for each gene of interest were normalized to their corresponding *Rps18* values. The comparative threshold cycle method was applied in the quantitative real-time PCR assay according to the 2^−ΔΔCt^ threshold cycle method. Each run was replicated at least three times.

### 2.6. Western Blot Analysis

Protein was extracted with RIPA (Solarbio, Beijing, China) with 1% PMSF (Solarbio, Beijing, China) and 1% protein phosphatase inhibitor (Huangxingbio, Beijing, China). Protein concentrations were determined by BCA protein assay kit (Solarbio, Beijing, China) and read at 562 nm. The same amounts of protein were fractionated by electrophoresis on a 10% SDS-PAGE gel, and electro-transferred to the PVDF membrane. Then, the protein was probed with antibodies specific to RPS18 (DF3679, Affinity, Jiangsu, China), phosphorylated(*p*)-Erk1/2(AF1015, Affinity, Jiangsu, China) and total(*t*)-Erk1/2 (AF0155, Affinity, Jiangsu, China), *p*-p38(AF4001, Affinity, Jiangsu, China) and *t*-p38 (AF6456, Affinity, Jiangsu, China), *p*-JNK (AF3318, Affinity, Jiangsu, China) and *t*-JNK (AF6318, Affinity, Jiangsu, China), *p*-Akt(AF0016, Affinity, Jiangsu, China) and *t*-Akt (AF6261, Affinity, Jiangsu, China), *p*-PKC(AF3197, Affinity, Jiangsu, China) and *t*-PKC (AF6197, Affinity, Jiangsu, China), Flag tag (T0003, Affinity, Jiangsu, China), RUNX2 (AF5186, Affinity, Jiangsu, China), COLI (14695-1-AP, Proteintech Group, Inc, Chicago, American), ALP (DF6225, Affinity, Jiangsu, China), *t*-AMPKα (5831, Cell Signaling Technology, Shanghai, China), *p*-AMPKα (50081, Cell Signaling Technology, Shanghai, China), MFN2 (DF8106, Affinity, Jiangsu, China), DRP1 (DF7037, Affinity, Jiangsu, China), OPA1 (DF8587, Affinity, Jiangsu, China). Image J Fiji (National Institutes of Health, Bethesda, MD, USA) was used to conduct semi-quantitative analysis by calculating the grey value of strips. The level of protein expression was normalized relative to the control, RPS18. The numbers below the strips are the results of the semi-quantitative assay which derived from at least three independent experiments. 

### 2.7. Staining and Activity of ALP 

ALP staining was performed after 7 days of osteo-induction. Cell cultures were washed three times with PBS and fixed with 4% paraformaldehyde for 5–10 min. ALP staining was performed using BCIP/NBT Alkaline Phosphatase Color Development Kit (Beyotime, Shanghai, China). Cells were stained with ALP detection solution and Alizarin red (Cyagen, Suzhou, Jiangsu, China) separately for over 30 min at room temperature, and then rinsed with PBS three times to remove excess staining. The quantification of ALP activity was processed using Alkaline Phosphatase Assay Kit (Beyotime, Shanghai, China) and monitored at 405 nm. Total protein was measured spectrophotometrically using a BCA protein assay Kit (Solarbio, Beijing, China) and read at 562 nm. ALP activity was normalized to the total protein content of the sample. 

### 2.8. Alizarin Red Staining and Quantification of Mineralized Osteoblast Cultures

Alizarin red staining was performed after 2–3 weeks of osteogenic induction. Cell cultures were washed three times with PBS and fixed with 4% paraformaldehyde for 5–10 min. Alizarin red staining was performed using alizarin red S solution (Solarbio, Beijing, China). For the quantification of mineralized osteoblast cultures, bound dye was dissolved with acetic acid and glycerol in a ratio of 5:1 and monitored at 405 nm wavelength.

### 2.9. XF24 Cell Mitochondria Stress Test

The oxygen consumption rate (OCR) and extracellular acidification rate (ECAR) was determined using Seahorse XFe24 Flux Analyzer (Agilent Technologies, Santa Clara, CA, USA). Transfected MC3T3-E1 were plated onto a Seahorse Xfe24 cell culture plate and osteogenic differentiation was induced after cell attachment overnight at a density of 1 × 10^5^ cells/well. One hour before the assay, the cells were washed twice with XF assay medium (31.66 mM NaCl + 10 mM glucose + 1 mM pyruvate + 2 mM glutamine) and were incubated in 500 μM assay medium at 37 °C in a non-CO_2_ incubator. Three mitochondrial inhibitors were diluted in assay medium and sequentially injected into each well during the measurements, oligomycin (an ATP synthase blocker, 1.5 μM), FCCP (an electron transport chain accelerator, 1 μM), Rotenone/Antimycin A (the inhibitors of mitochondrial respiratory chain complex I and III, 0.5 μM). Indicators of mitochondria respiratory function include basal respiration, proton leak (oligomycin response—rotenone and antimycin A response), ATP-linked respiration (basal respiration—oligomycin response), spare capacity (maximal respiration—basal respiration), maximal respiration (FCCP response—rotenone and antimycin A response) and non-mitochondrial oxygen consumption (rotenone and antimycin A response), and the ECAR of glycolytic reserve (oligomycin response—basal glycolysis) was calculated for the evaluation of glycolytic function.

### 2.10. Mitochondrial Staining

For mitochondrial imaging, MitoTracker Red CMXRos (Solarbio, Beijing, China) was used according to the instructions. The adherent cells on coverslips were incubated with 100 nM MitoTracker red for 30 min. Then, the nuclei were stained with fluorescent mounting medium (ZLI-9556, ZSGB-BIO, Beijing, China) on a glass slide. The morphology of mitochondria was observed through TCS-SP8 DIVE (Leica, Wetzlar, Germany) and mitochondria 2D analysis was conducted by Image J Fiji according to the literature [22].

### 2.11. Culture of Calvarias

Calvarias were dissected from 3-day-old mice and were cut along the middle line into two parts. Two parts supported by a metal mesh were cultured in basal medium with 1 mg/L albumin bovine V (A8020, Solarbio, Beijing, China), 2 mM glutamine (CG5801, Coolaber, Beijing, China), 50 ng/mL ascorbic acid (CA2261, Coolaber, Beijing, China), 10 mM β-glycerophosphate disodium (CG5821, Coolaber, Beijing, China), 10% FBS and antibiotics in a humidified atmosphere of 5% CO_2_ at 37 °C. The culture method was according to the literature [23]. One part of calvarias was treated with DMSO (Solarbio, Beijing, China) as a control and another part was treated with U0126 (S1102, Selleck Chemicals, Shanghai, China) or Compound C (171261, Merck, Shanghai, China).

### 2.12. Micro-CT Scan and Analysis

The calvarias cultured for 7 days were fixed with 4% paraformaldehyde (Solarbio, Beijing, China) at 4 °C for 2 days. The samples were scanned with micro-CT (vivaCT 80, SCANCO Medical AG, Zurich, Switzerland) under the condition of 70 kVp, 114 A, 8 W. Two-dimensional images were used to generate three-dimensional reconstructions. Mimics 20.0 (Materialise, Leuven, Belgium) was used to obtain two-dimensional and three-dimensional images with the same segmentation threshold. 

### 2.13. H&E Staining

After 7 days of culture, the calvarias were decalcified, dehydrated and paraffin-embedded. According to H&E Staining Kit instructions (Solarbio, Beijing, China), the slices were dried in the oven overnight and dewaxed in xylene for 5 min. Then, rehydration was taken in gradient concentration of ethanol. Hematoxylin staining, differentiation liquid differentiation, eosin immersion, dehydration, transparent and neutral gum sealing were performed. 

### 2.14. Statistical Analysis 

Data were evaluated in GraphPad prism, version 6.0 (GraphPad Software, San Diego, CA, USA). Results are as shown as the mean ± SD. Statistics were assessed using *t*-test, assuming double-sided independent variance and *p* values were significant at * *p* < 0.05, ** *p* < 0.01, ****p* < 0.001 and **** *p* < 0.0001.

## 3. Results

### 3.1. Clinical Findings

Two sporadic cases of Crouzon syndrome were recruited in this study. 

Patient A, a three-year-old male, presented abnormal head shape and exophthalmos at one month and was diagnosed with Crouzon syndrome at forty days after birth. Cranial vault reconstruction was undertaken when he was two months old. Midfacial hypoplasia became increasingly serious after one year. Obstructive sleep apnea syndrome and laryngitis forced him to accept endotracheal intubation at sixteen months old. Clinical examinations revealed the premature closure of the bilateral coronal suture, acrobrachycephaly, midfacial hypoplasia with anterior crossbite, large and shallow orbits, exophthalmos and small stature but no syndactyly (Figure 1A). 

Patient B, a four-year-old female, had difficulty in breathing and presented wheezing and thus she could not lie flat to sleep after birth. Cranial vault reconstruction was performed at four months old. The eye protrusion became increasingly serious along with her growth, and the eyelids could not be completely closed when sleeping. Clinical examinations revealed the acrobrachycephaly, exophthalmos, shallow orbits, orbital hypertelorism, and hyperemia of the conjunctiva but no syndactyly. She also manifested other typical symptoms of the Crouzon syndrome of serious midfacial hypoplasia with anterior crossbite, and the severe crowding of upper dentition (Figure 1A).

### 3.2. Identified Mutation and Pathogenic Assessments

The genetic assessments revealed a heterozygous missense mutation, FGFR2 exon 8 c.1024T>C, p.Cys342Arg present in the disease phenotype, and absent in 50 healthy controls (Figure 1B). The prediction results of the functional effect of this human non-synonymous single-nucleotide polymorphisms (nsSNP) are shown in Appendix A that was considered intolerant and harmful. This FGFR2 mutation (p.Cys342Arg) resulted in the replacement of cysteine with arginine at codon 342 in the Ig-III domain. The mutational spot was highly conserved among all tested species (Figure 1C). Protein structure simulation showed that there were reduced helices in the secondary structure near the mutation site circled by red boxes (Figure 1D). A disulfide bond was also disconnected which was originally formed by cysteine at codon 342 and cysteine at codon 278, which made the spatial structure unstable and exposed more active sites (Figure 1E).

### 3.3. Mutation of FGFR2 Promotes Osteogenic Differentiation of Pre-Osteoblasts and Overactivation of Erk1/2 Pathway Influences Osteogenesis 

The structure of FGFR2 with p.Cys342Arg mutation was shown in Figure 2A. The site of FGFR2 mutation (p.Cys342Arg) was indicated by red arrow. Changes in cell proliferation and osteogenesis were detected after wild-type and mutant FGFR2 transfection for the indicated period. The CCK8 result showed the more exuberant proliferation of MC3T3-E1 cells in the mutant (MT) group than in the wild-type (WT) group (Figure 2B). qRT-PCR and Western blot showed the enhanced expression of the early osteogenic genes of *Runx2*, *Alp*, *ColIα2*, and late osteogenic genes of *Opn* and *Ocn* in the MT group (Figure 2C). The ALP staining, ALP activity, and alizarin red staining and quantification results show that an increased number of crystal violet-staining cells, ALP activity, mineralized nodules and absorbance of alizarin red were found in the MT group than in WT group (Figure 2D). All evidence proved that FGFR2 mutation (p.Cys342Arg) could enhance osteogenesis.

To detect the alteration of the cardinal signaling pathway affected by this mutation in the osteogenesis process, FGF/FGFR2 downstream signaling pathways were analyzed by Western blot. There was no significant change in the expression of *t*-FGFR2 and *t*-Erk1/2 in the WT and MT group, but the expression of *p*-FGFR2 and *p*-Erk1/2 was markedly upregulated in the MT group. Meanwhile, there were no obvious differences in the expression of JNK, p38, Akt, PKC and their phosphorylation between WT and MT groups (Figure 2E). The above results suggest that the FGFR2 mutation (p.Cys342Arg) brings about the constitutive activation of the FGF/FGFR2-Erk1/2 MAPK signaling pathway and that the classical Erk1/2 MAPK pathway is involved in the positive regulation on the proliferation and osteogenic differentiation of osteoblast precursors.

### 3.4. siRNA-Induced Knockdown of FGFR2 Expression Inhibits Osteogenic Differentiation and Erk1/2 Signaling Pathway In Vitro

To further investigate the influence of FGFR2 on cell proliferation and osteogenic differentiation, we constructed siRNA by specifically knocking down FGFR2. qRT-PCR was used to verify that the FGFR2 mRNA level was decreased in the siFGFR2 group compared with that in the negative control (siNC) group (Figure 3A). The transfection efficiency of siRNA was detected by fluorescence microscopy (Appendix A).

CCK-8 showed that the knockdown of FGFR2 delayed the proliferative activity of MC3T3-E1 cell lines (Figure 3B). qRT-PCR and Western blot demonstrated that the expression of osteoblast marker genes, *Alp*, *Runx2*, *ColI*, *Opn* and *Ocn,* was decreased in the siFGFR2 group (Figure 3C). Meanwhile, the knockdown of FGFR2 weakened the ALP activity and decreased mineralization of MC3T3-E1 cells (Figure 3D). 

Western blot analysis showed that the expression of the *t*-FGFR2, *p*-FGFR2, and *p*-Erk1/2 level decreased in siFGFR2 group. Similarly, there were no significant differences in JNK, p38, Akt, PKC, and their corresponding phosphorylation levels between siNC and sFGFR2 groups (Figure 3E).

All of the above results determined once again that the FGFR2 mutation (p.Cys342Arg) was a constitutive activation mutation and that FGF/FGFR2-Erk1/2 MAPK played an positive role in cell proliferation and osteogenic differentiation.

### 3.5. Domain Negative FGFR2 Yields the Weakened Osteogenic Differentiation through Erk1/2 Signaling Pathway 

In order to prove that the constitutive activation mutation does excessively phosphorylate the intracellular tyrosine kinase domain to initiate the Erk1/2 signaling cascade, another plasmid was constructed that expressed domain negative (DN) FGFR2. DN FGFR2 retained the mutation and did not have the intracellular tyrosine kinase domain like an antenna fixed on the cell, so that it could sense the external signals but lose intracellular signal transduction function.

The CCK8 result shows that the proliferative capability of MC3T3-E1 cells in the DN group was significantly inhibited (Figure 4A). qRT-PCR and Western blot analysis showed a lower expression of osteogenic differentiation markers at both transcriptional and translational levels in the DN group (Figure 4B). ALP staining showed a decreased formation of ALP-positive colonies in the DN group. Alizarin red staining showed that DN FGFR2 reduced mineralization. The quantitative results of ALP activity and mineralized nodules were also consistent with the histochemistry results (Figure 4C).

Meanwhile, Western blot analysis showed that the expression of *t*-FGFR2 and *p*-FGFR2 was decreased in the DN group compared with the MT group. The decreased expression of FGFR2 was probably because of the FGFR2 antibody corresponding to amino acid residues V754-Y779 which was not expressed in DN FGFR2 plasmid. In addition, *p*-Erk1/2 was obviously inhibited in the DN group (Figure 4D).

### 3.6. Decreased Osteogenic Differentiation Activity after Using Erk1/2 Inhibitor U0126 

To determine whether the Erk1/2 MAPK signaling pathway was involved in proliferation and osteogenic differentiation, we used MEK1/2 inhibitor U0126 (S1102, Selleck Chemicals, Shanghai, China) against Erk1/2. DMSO (Solarbio, Beijing, China) was added as the control group.

As shown in the Western blot analysis in Figure 4E, *p*-Erk1/2 was inhibited after being treated with U0126. As a result, the CCK-8 assay showed that cell proliferation was decreased (Figure 4F). qRT-PCR and Western blot analysis demonstrated that the expressions of osteoblast markers *Alp*, *Runx2*, *ColI*, *Opn* and *Ocn* were reduced by the U0126 treatment (Figure 4E,G). ALP activity and mineral content were also decreased after U0126 treatment (Figure 4H). It was verified again that Erk1/2 MAPK had a considerably positive role in proliferation and osteogenic differentiation. 

### 3.7. Activation of AMPK-Erk1/2 Pathway Contributes to Dysregulated Osteogenesis by FGFR2 p.Cys342Arg

As the key protein of energy metabolism, the expression of AMPK was detected. The Western blot showed that the expression of *t*-AMPK and *p*-AMPK was upregulated in the MT group compared with the WT group, and the expressions of *t*-AMPK and *p*-AMPK were downregulated in the siFGFR2 group compared with the siNC group (Figure 5A). This demonstrates that the level of FGFR phosphorylation could affect the expression of AMPK. Based on this finding, we examined whether the activation of AMPK might mediate the osteogenic differentiation of MC3T3-E1. An inhibitor of AMPK signaling Compound C (171261, Merck, Shanghai, China) blockade assay showed that there was less ALP activity and decreased mineral content (Figure 5B). Meanwhile, the qRT-PCR and Western blot showed that osteogenic genes including *Alp*, *Runx2*, *ColI*, *Opn* and *Ocn* expression were reduced by the treatment of Compound C (Figure 5C). Therefore, the activation of AMPK was necessary for FGF/FGFR-Erk1/2-regulated osteogenesis. 

To further confirm the relationship between the AMPK and Erk1/2 pathways in the osteo-differentiation process, Compound C and U0126 were adopted. Figure 5D showed that the expression of *p*-Erk effectively decreased after the treatment of Compound C. However, the level of *p*-AMPK did not show a significant change after the U0126 treatment, indicating that Erk1/2 was the downstream target of AMPK.

### 3.8. Augmentation of Mitochondria Respiration Function Impacted by Overactivated FGF/FGFR-AMPK

The capacity of ATP production and cell respiration are some of the indicators of normal mitochondrial function regulated by AMPK [24,25]. Mitochondrial stress test by seahorse assay was performed to measure OCR of MC3T3-E1 to evaluate mitochondrial respiratory capacity during osteogenic differentiation. 

Compared with the WT group (128.01 ± 25.16 pmol/min), there was an increase in the basal respiration of the MT group (189.22 ± 24.96 pmol/min). The basal respiration consisted of two parts, namely ATP-linked respiration and proton leakage. ATP-linked respiration was enhanced in the MT group (169.41 ± 23.46 pmol/min) compared to the WT group (115.52 ± 22.90 pmol/min). The maximal respiratory capacity stimulated by the injection of FCCP was significantly higher in the MT group than that in the WT group, 547.61 ± 87.60 pmol/min vs. 350.64 ± 109.08 pmol/min. The same was true for spare capacity, which was 358.39 ± 64.05 pmol/min in MT group and 222.63 ± 84.41 pmol/min in WT group (Figure 6A,B).

After the knockdown of FGFR2 by siRNA in MC3T3-E1, the maximal respiration (354.50 ± 95.67 pmol/min) and spare capacity (209.24 ± 78.07 pmol/min) were about the half of maximal respiration (605.74 ± 161.03 pmol/min) and spare capacity (398.69 ± 135.26 pmol/min) in the siNC group. Additionally, the basal respiration was obviously reduced after the knockdown of FGFR2 (145.26 ± 18.28 pmol/min) compared with the control (207.06 ± 27.75 pmol/min). The respiration for the ATP production of 130.59 ± 15.05 pmol/min in siFGFR2 group was less than 182.14 ± 23.70 pmol/min in the siNC group (Figure 6C,D).

To verify whether OCR was impacted by the activation of AMPK, the AMPK inhibitor Compound C was applied, and it was revealed that the indicators of cell respiratory function were decayed—especially maximal OCR (20.70 ± 20.95 pmol/min) and spare respiratory capacity (86.59 ± 21.92 pmol/min), compared to 150.93 ± 32.84 pmol/min of maximal OCR and 239.42 ± 35.41 pmol/min of spare respiratory capacity for the control. There was a slight reduction in basal respiration of 65.90 ± 10.96 pmol/min in Compound C group vs. 88.49 ± 10.39 pmol/min in the control group, in addition to a slight reduction in ATP production of 58.22 ± 10.80 pmol/min in the Compound C group vs. 83.90 ± 6.50 pmol/min in the control group (Figure 6E,F). 

MC3T3-E1 with FGFR2 p.Cys342Arg expanded its glycolytic reserve to satisfy the demand for osteo-differentiation because of the upregulation of AMPK, even though these degrees of change were not as obvious as that of the oxidative phosphorylation (OXPHOS) (Appendix A).

### 3.9. The Effects of FGF/FGFR2-AMPK Axis on Mitochondrial Morphology 

To examine the effect of FGFR2 mutation p.Cys342Arg on mitochondrial morphology in MC3T3-E1, Mito-tracker red staining was used to compare the mitochondrial morphology and the expression of genes related to mitochondrial morphology was examined by qRT-PCR and Western blot. 

In the MT group, the mitochondria closely surrounded the nucleus and overall showed a more punctate and shorter stick; however, the mitochondria in the WT group presented a widely reticular formation distributed throughout the cytoplasm (Figure 7A). The number of branches, branch joints and the branch length of mitochondria in the MT group were smaller than that in the WT group. These changes in morphology were accompanied by a lower mRNA expression of mitochondria-fusion-related *Mfn2* and *Opa1*, but a higher mRNA expression of fission-related molecule *Drp1* in the MT group (Figure 7B). The Western blot confirmed that MFN2 and OPA1 were down-regulated and DRP1 was upregulated at the translational level in the MT group (Figure 7B).

In the siFGFR2 group, the interconnected reticulate mitochondria were extensively scattered throughout the cytoplasm, however, in the siNC group, more dot-shaped or club-shaped mitochondria spread in the cytoplasm (Figure 7C). Mitochondrial 2D analysis by Image J Fiji showed the number of branches, whilst the branch joints and branch length were larger than those in the siNC group after the knockdown of FGFR2. Additionally, in the siFGFR2 group, the expressions of *Mfn2* and *Opa1* were increased, and *Drp1* was decreased at both transcriptional and translational levels (Figure 7D).

Then, Compound C was applied to inhibit AMPK activation to detect the modulation of mitochondrial morphology. It was found that mitochondria in the MT group which should have been rod-shaped returned to a wire-mesh-like structure (Figure 7E). Meanwhile, the level of fusion-related molecules MFN2 and OPA1 was upregulated and fission-related molecule DRP1 was downregulated after treatment with Compound C (Figure 7F).

### 3.10. Inhibitors Attenuated the Closure of Coronal Sutures of Cultured Calvarias 

It was shown in Figure 8A that there was an obvious linear gap between the frontal and parietal bone that was not completely closed after the treatment with U0126 and Compound C. In the control group, the coronal suture region was filled with bone tissue. As shown by the section selected at the corresponding position of micro-CT (Figure 8B), there were gaps between the osteogenic fronts of frontal and parietal bone in inhibitor groups. The osseous cross-linking in the coronal suture was closer in the control groups. 

As revealed by H & E staining (Figure 8B), the synostosis between the frontal and parietal bone presented more serrated in control. However, there was a decreased overlapping region after the treatment with U0126 and Compound C. Both in the inhibitor group, the coronal sutures were in a non-dense partially fused state.

## 4. Discussion

In this study, we fully investigated the FGFR2 mutation p.Cys342Arg detected in two Chinese Crouzon syndrome patients and revealed the novel mechanism whereby FGFR2 p.Cys342Arg -AMPK-Erk1/2 axis regulates osteogenesis by enhancing the mitochondrial metabolism (Figure 8C). Inhibitors targeting either AMPK or Erk1/2 have potential as alternative treatment approaches for intervening in the Crouzon syndrome disease process in the future.

To the best of our knowledge, through gene detection and bioinformatics analysis, some infrequent cases have reported FGFR2 mutation p.Cys342Arg as the genetic cause of Crouzon syndrome or other syndromes with craniosynostosis, such as Pfeiffer and Jackson–Weiss. However, the influence of this mutation on cell proliferation and osteogenic differentiation and its underlying mechanism might not have been fully investigated [3,20,26,27]. Previous research on other similar variants showed that the Erk1/2 and p38 signaling pathway jointly contributed to skull abnormalities caused by the FGFR2 mutation (p.Pro253Arg) in Apert syndrome mice. Additionally, the Erk1/2 and PKC signaling pathways played an essential role in promoting osteogenic differentiation of murine mesenchymal stem cells with FGFR2 mutation (p.Ser252Trp) [17,19]. In addition, previous studies found the upregulation of *p*-Erk in the osteogenic fronts of cranial sutures in mouse models with FGFR2 mutation (p.Cys342Tyr) [28]. According to our results, the disruption of the osteogenic differentiation by the Erk1/2 signaling pathway could be one of the reasons for the overgrowth of bone tissues in the primary patient sutures and resulting in craniosynostosis in the genetic background of FGFR2 mutation p.Cys342Arg. In vitro experiments using MC3T3-E1 treated with U0126, the expression of osteogenic genes was decreased and osteogenic differentiation was inhibited. The results of the delayed closure of cranial sutures cultured in vitro showed that the Erk1/2 signal was a core pathway which has been extensively studied in past decades and that could be used as a potential therapeutic target [18,19].

Additionally, we noticed the constitutive activation of FGFR2 p.Cys342Arg or knockdown of FGFR2 affected the expression and phosphorylation of AMPK, and the inhibition of AMPK further reduced the osteogenic differentiation of MC3T3-E1 and delayed closure of the cranial suture. Therefore, AMPK could be a potential therapeutic target for syndromic craniosynostosis. Although the positive role of AMPK in osteogenesis has been newly reported [29,30,31], the correlation between Erk1/2 and the AMPK pathway in FGFR2 mutation-induced osteogenesis remain unclear. Thus, we adopted two inhibitors to confirm that AMPK with positive effects was the upstream signal of Erk1/2, which made the role of AMPK more central. However, the results differed from those of previous studies reporting that the AMPK pathway was downstream of Erk1/2 signaling during nanomaterial-induced osteogenesis [32] and the AMPK pathway negatively regulated the Erk1/2 signal by suppressing B-raf on melanoma [33]. 

Additionally, we found the AMPK-mediated mitochondrial respiratory function to orchestrate cranial suture fusion. The production of ATP is closely related to the oxygen consumption of cell respiration which could be promoted by AMPK [25,34]. The constitutive activation of the FGFR2 (p.Cys342Arg) mutation yielded the overexpression and activation of AMPK, and greatly augmented the capacity of ATP production (ATP-linked respiration) and the adaptability of mitochondrial respiratory function (spare respiratory capacity). These results indicate that osteogenic activities have become increasingly dependent on ATP produced by OXPHOS. Enhancing OXPHOS by the activation of AMPK could accelerate osteogenic differentiation [24,34,35]. 

Mitochondria are highly dynamic organelles continuously in the transition of fusion and fission to generate an interconnected network responding to energy stress [36]. Some studies reported that excessive mitochondrial fragmentation in inflammation with ROS-induced oxidative stress inhibited osteogenesis [37,38]. However, we captured mitochondrial structures with an increasing fission rate influenced by FGFR2 p.Cys342Arg to accelerate osteo-differentiation. The knockdown of FGFR2 and inhibiting AMPK activation converted fission into fusion. Combined with previous findings that inhibiting ATP synthesis could be compensated by provoking AMPK activation and then inducing mitochondrial fission [25,36,39], we have reason to believe that increasing mitochondrion division to enrich mitochondrial mass mediated by AMPK is certainly important to meet the energy needed for the enhanced osteogenic activity caused by the continuous activation of the FGF/FGFR2 pathway.

In addition to positively regulating Erk1/2, AMPK can also affect the energy metabolism oxidative phosphorylation involved in mitochondrial dynamics, making it a potential therapeutic target for craniosynostosis.

## 5. Conclusions

In this study, we detected the FGFR2 mutation (p.Cys342Arg) with no experiment proof ever in two Chinese sporadic patients with Crouzon syndrome. The constitutive activation mutation p.Cys342Arg in FGFR2 enhances osteogenesis via a vitally significant FGF/FGFR2-AMPK-Erk1/2 MAPK signaling pathway which is mitochondrial metabolism-mediated. AMPK and Erk1/2 inhibitors could be potential remedies for Crouzon syndrome in the future.

## Figures and Tables

**Figure 1 cells-11-03129-f001:**
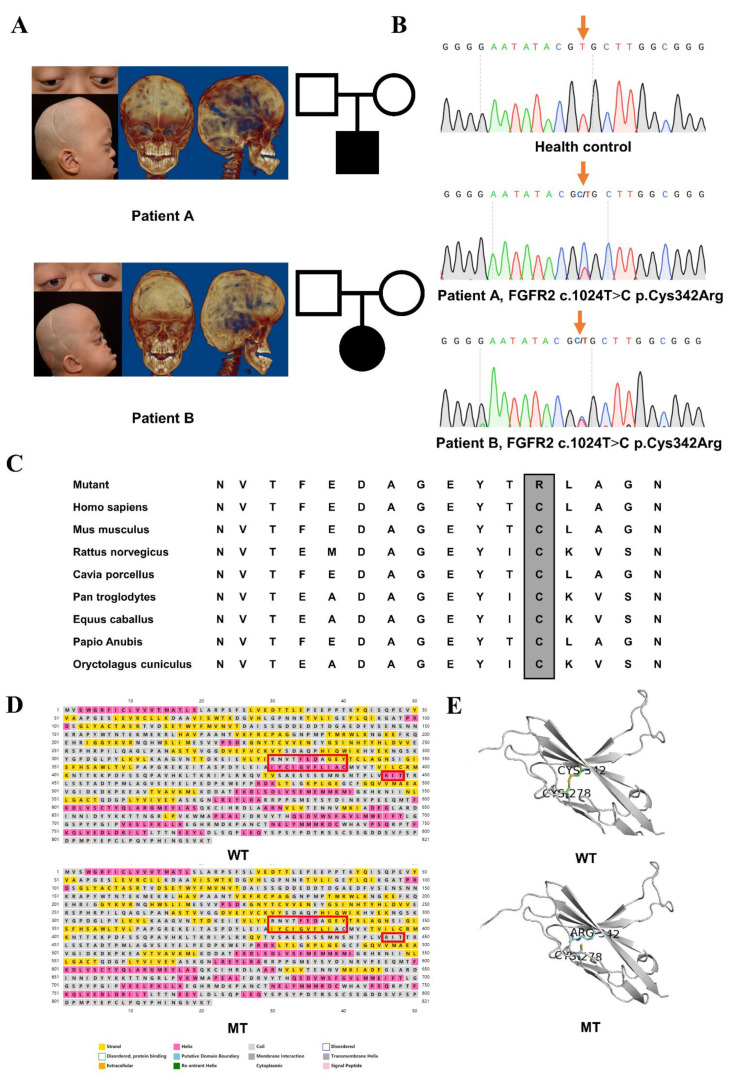
Clinical findings, identified mutation and pathogenic assessments: (**A**) Digital photos and CT of two sporadic Chinese children with Crouzon syndrome after cranial vault reconstruction. Both of them suffered from the premature closure of the bilateral coronal sutures, acrobrachycephaly, exophthalmos, midfacial hypoplasia and crossbite. (**B**) Sequencing chromatograms of the identified FGFR2 c.1024T>C mutation which was marked by an orange arrow and wild-type sequence. (**C**) Residue at codon 342, indicated by the frame, was evolutionarily conserved among all tested species. (**D**) Helices were reduced in p.Cys342Arg mutant FGFR2 circled by red boxes. (**E**) A disulfide bond which was originally formed by cysteine at codon 342 and cysteine at codon 278 was disconnected because of p.Cys342Arg.

**Figure 2 cells-11-03129-f002:**
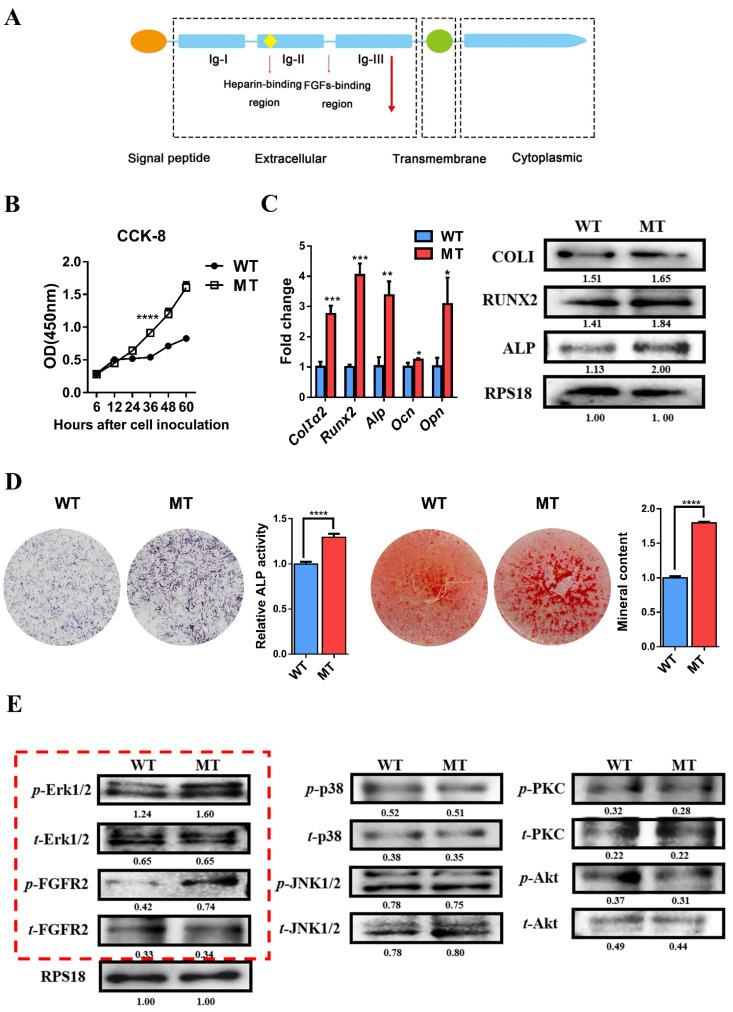
Constitutive activation of FGFR2 (p.Cys342Arg) enhanced osteogenesis via Erk1/2 MAPK signaling pathway: (**A**) Schematic representation of the relative linear location in which the FGFR2 mutation is identified as illustrated by the large red arrow shown in the context of the protein structure. (**B**) CCK-8 assay was carried out to assess cell proliferation. Proliferation of MC3T3-E1 cells was more active in the MT group. (**C**) Relative expressions of osteogenic marker measured by qPCR. The expressions of *Alp*, *ColIα2*, *Runx2*, *Opn* and *Ocn* mRNA in differentiated MC3T3-e1 were remarkably increased in the MT group. Western blot analysis showed that the level of ALP, COLI and RUNX2 were also increased in the MT group. (**D**) ALP staining, alizarin red staining and quantitative tests. ALP staining showed increased crystal violet-staining cells in the MT group compared with the WT group. Quantitative experiment demonstrated that ALP activity is more active in the MT group. Alizarin red staining and quantitative test showed there were more mineralized nodules and mineral content in the MT group than in the WT group. (**E**) Western blot analysis demonstrated that the levels of *p*-FGFR2 and *p*-Erk1/2 were increased in the MT group. There was no significant change in the expression of key proteins in other downstream pathways. The western blot results of FGF/FGFR2-Erk1/2 was circled by the red frame. *p* values were significant at * *p* < 0.05, ** *p* < 0.01, *** *p* < 0.001 and **** *p* < 0.0001.

**Figure 3 cells-11-03129-f003:**
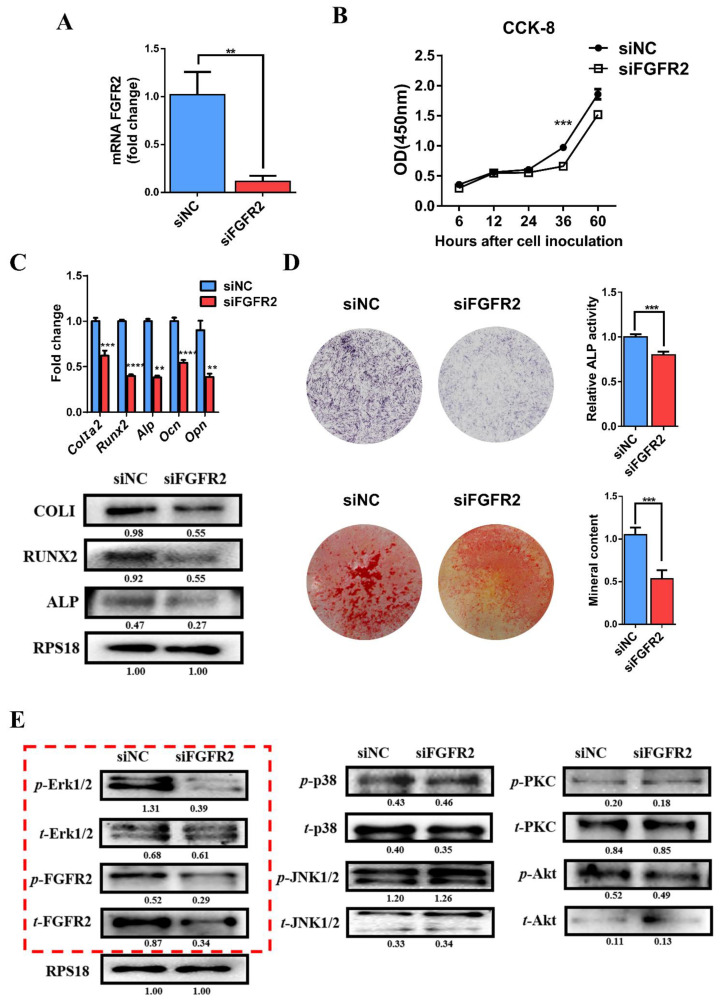
Knockdown of FGFR2 induced by siRNA-inhibited osteogenic differentiation by decreasing Erk1/2 MAPK activation: (**A**) qRT-PCR assay showed mRNA FGFR2 expression after transfection with siRNA against FGFR2. (**B**) CCK-8 assays were carried out to assess cell proliferation. Knockdown of FGFR2 inhibited the proliferation of MC3T3-E1 cells. (**C**) Relative expressions of osteogenic marker measured by qRT-PCR and Western blot. The expression of *Alp*, *ColIα2*, *Runx2*, *Opn* and *Ocn* mRNA in differentiated MC3T3-E1 were remarkably decreased in siFGFR2 group. The level of ALP, COLI, RUNX2 protein were also decreased in siFGFR2 group. (**D**) ALP staining and quantitation of ALP activity showed that the ALP activity was decreased in the siFGFR2 group compared with the control. Alizarin red staining and quantitative test demonstrated there was less mineral content in the siFGFR2 group. (**E**) Western blot analysis demonstrated that the level of *t*-FGFR2, *p*-FGFR2 and *p*-Erk1/2 was decreased in the siFGFR2 group. There was no significant change in the expression of key proteins in other downstream pathways. The western blot results of FGF/FGFR2-Erk1/2 was circled by the red frame. *p* values were significant at ** *p* < 0.01, *** *p* < 0.001 and **** *p* < 0.0001.

**Figure 4 cells-11-03129-f004:**
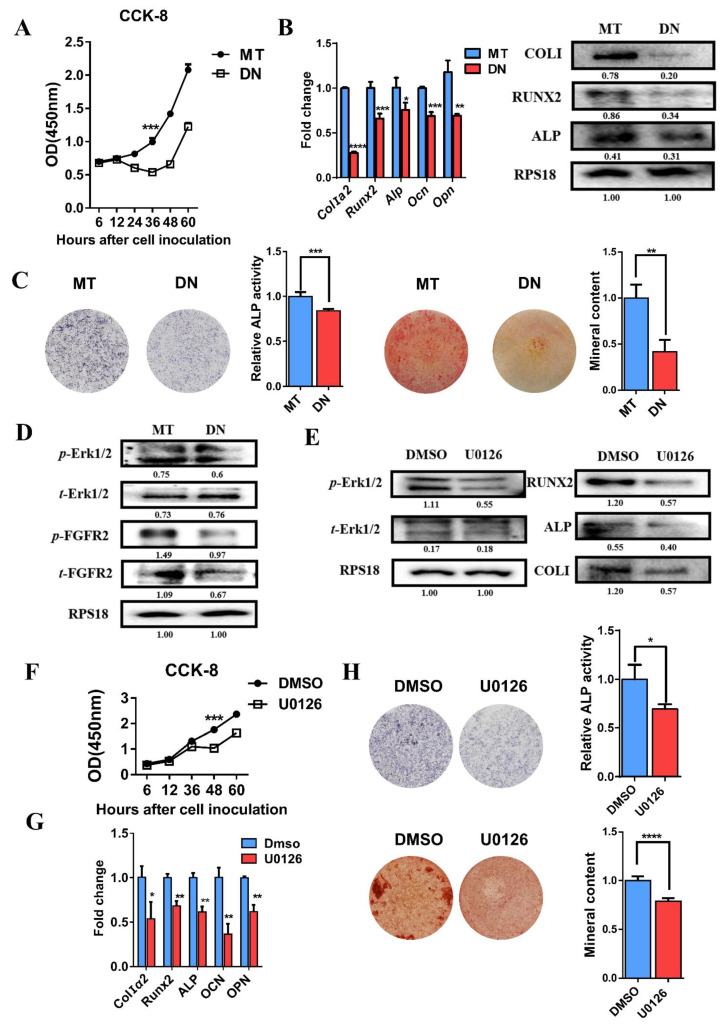
Domain negative FGFR2 weakened osteogenesis through the Erk1/2 signaling pathway: (**A**) CCK-8 assays showed the proliferation of MC3T3-E1 cells was inhibited in the DN FGFR2 group compared with the MT group. (**B**) Relative expressions of osteogenic genes such as *ColIα2*, *Runx2*, *Alp*, *Ocn* and *Opn* were decreased in the DN group compared with the MT group measured by qRT-PCR. The level of ALP, COLI and RUNX2 was inhibited in the DN group, as detected by Western blot. (**C**) ALP staining and quantification of the ALP activity of MC3T3-E1 was decreased in the DN group. Alizarin red staining and the quantification of mineral content demonstrated that the mineral content of MC3T3-E1 was significantly reduced in the DN group. (**D**) Western blot showed that the expressions of *t*-FGFR2, *p*-FGFER2 and *p*-Erk1/2 were remarkably decreased in the DN group. (**E**) Alterations after using U0126 proved the positive role of Erk1/2 MAPK on osteogenesis. Western blot demonstrated that the Erk1/2 MAPK pathway was inhibited by U0126 and the levels of osteogenic marker, COLI, RUNX2 and ALP were decreased. (**F**) MEK inhibitor, U0126, suppressed the proliferation of MC3T3-E1 cells. (**G**) The expressions of *Alp*, *ColIα2*, *Runx2*, *Opn* and *Ocn* mRNA detected by qRT-PCR were reduced by using U0126. (**H**) ALP staining, alizarin red staining and quantification test demonstrated that both ALP activity and mineralization were decreased after the Erk1/2 MAPK pathway was inhibited. *p* values were significant at * *p* < 0.05, ** *p* < 0.01, *** *p* < 0.001 and **** *p* < 0.0001.

**Figure 5 cells-11-03129-f005:**
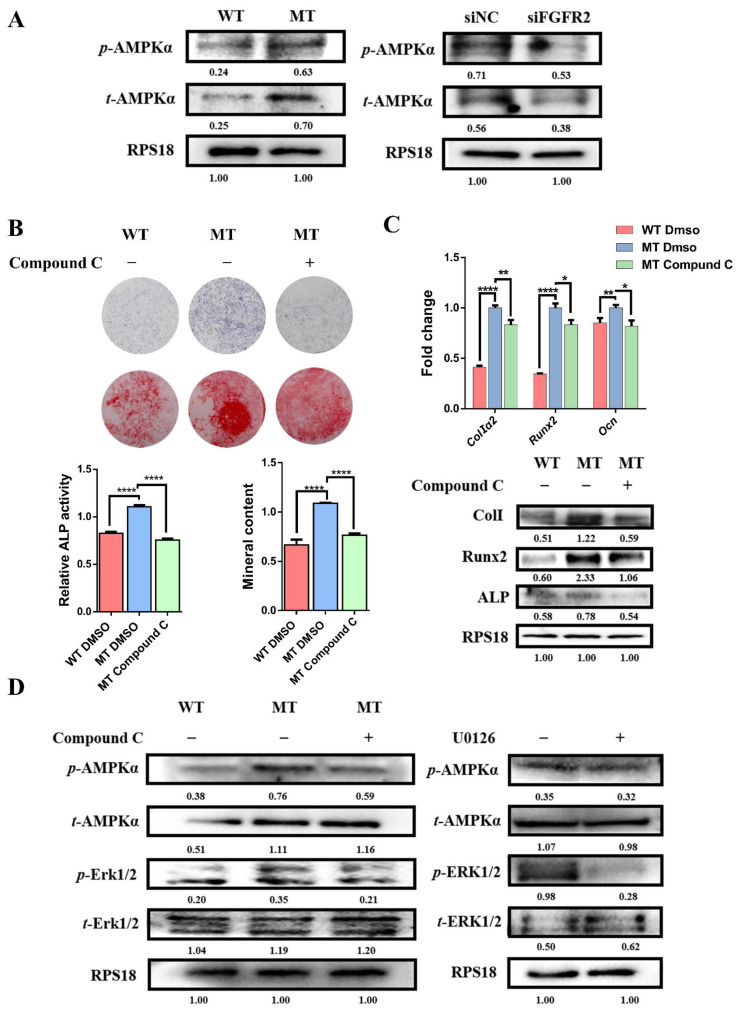
The activation of the AMPK-Erk pathway contributed to dysregulating osteogenesis by FGFR2 p.Cys342Arg: (**A**) Western blot analysis showed that the expression of *t*-AMPK and *p*-AMPK were upregulated in the MT group compared with the WT group and the expression of *t*-AMPK and *p*-AMPK were downregulated in the siFGFR2 group compared with the control. (**B**) ALP staining, alizarin red staining and quantification test demonstrated that both ALP activity and mineralization were decreased after the AMPK pathway was inhibited. (**C**) The expressions of osteogenic markers *Alp*, *ColI*, *Runx2* and *Ocn* were reduced after being treated with Compound C, as determined by qRT-PCR and Western blot. (**D**) Western blot analysis showed that the level of *p*-Erk was effectively inhibited after being treated with Compound C. However, the level of *p*-AMPK had no significant change after the U0126 treatment. *p* values were significant at * *p* < 0.05, ** *p* < 0.01 and **** *p* < 0.0001.

**Figure 6 cells-11-03129-f006:**
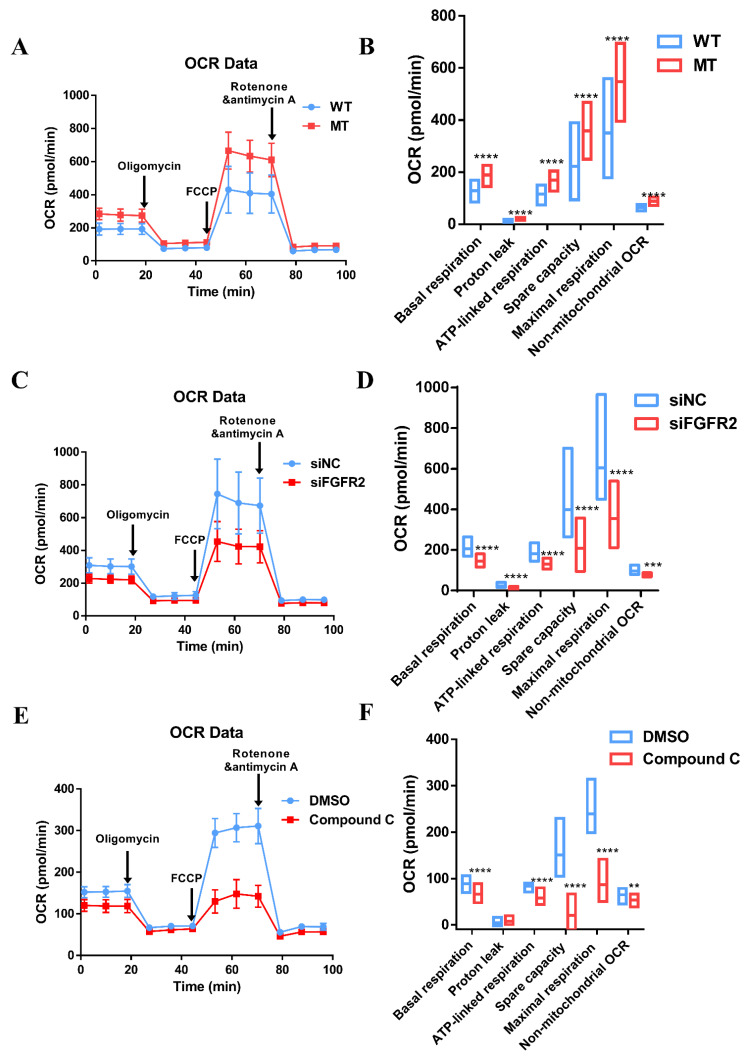
Mitochondria respiratory dysfunction influenced by the FGFR/FGFR2-AMPK pathway: (**A**) Traces of OCR in the WT group and MT group, respectively, in response to oligomycin, FCCP, and rotenone and antimycin A. (**B**) All parameters of cell respiratory function in the MT group were enhanced to varying degrees. The maximal respiratory capacity was significantly higher in the MT group than that in the WT group—547.61 ± 87.60 pmol/min vs. 350.64 ± 109.08 pmol/min, respectively. The same was true for spare capacity, with 358.39 ± 64.05 pmol/min in the MT group and 222.63 ± 84.41 pmol/min in the WT group. (**C**) Traces of OCR in the siNC group and siFGFR2 group. (**D**) After the knockdown of FGFR2, a completely opposite trend revealed itself. The maximal respiration (354.50 ± 95.67 pmol/min) and spare capacity (209.24 ± 78.07 pmol/min) were about the half of maximal respiration (605.74 ± 161.03 pmol/min) and spare capacity (398.69 ± 135.26 pmol/min) in the siNC group. (**E**) OCR traces were then treated with Compound C, an AMPK inhibitor. DMSO was added as the control. (**F**) Indicators of cell respiratory function were decayed, especially maximal OCR (20.70 ± 20.95 pmol/min) and spare respiratory capacity (86.59 ± 21.92 pmol/min), compared to 150.93 ± 32.84 pmol/min of maximal OCR and 239.42 ± 35.41 pmol/min of spare respiratory capacity in the control. *p* values were significant at ** *p* < 0.01, *** *p* < 0.001 and **** *p* < 0.0001.

**Figure 7 cells-11-03129-f007:**
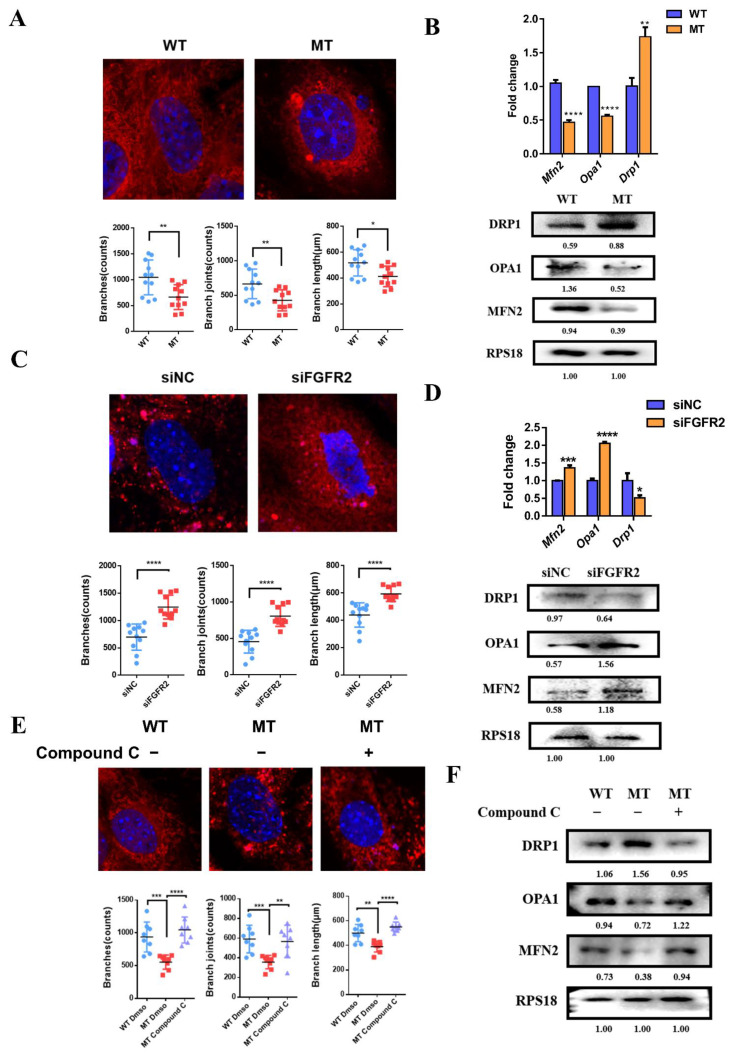
Mitochondrial dynamics participate in osteogenesis mediated by FGFR/FGFR2-AMPK pathway: (**A**) Mito-tracker red and mitochondria 2D analysis showed more fragmented or punctate mitochondria with fewer and shorter branches in the MT group, while there were widely reticular mitochondria with longer branches in the WT group. (**B**) qRT-PCR and Western blot analysis demonstrated that the expressions of mitochondrial-fusion-related factors *Mfn2* and *Opa1* were downregulated, however, the expression of mitochondrial-fission-related factor *Drp1* was upregulated in the MT group compared to the WT group. (**C**) Mito-tracker red and mitochondria 2D analysis showed a filamentous network of mitochondria with more and longer branches widely spread in the cytoplasm in the siFGFR2 group. (**D**) qPCR and Western blot analysis showed that the expression of fusion-related genes *Mfn2* and *Opa1* was increased and the expression of fission-related gene *Drp1* was decreased in the siFGFR2 group. (**E**) After treatment with Compound C, the morphology of the mitochondria which should have tended to split adopted a fused reticular structure with more and longer branches, as shown by Mito-tracker red and mitochondria 2D analysis. (**F**) Western blot analysis showed the level of MFN2 and OPA1 was increased and then level of DRP1 was decreased, as induced by Compound C. *p* values were significant at * *p* < 0.05, ** *p* < 0.01, *** *p* < 0.001 and **** *p* < 0.0001.

**Figure 8 cells-11-03129-f008:**
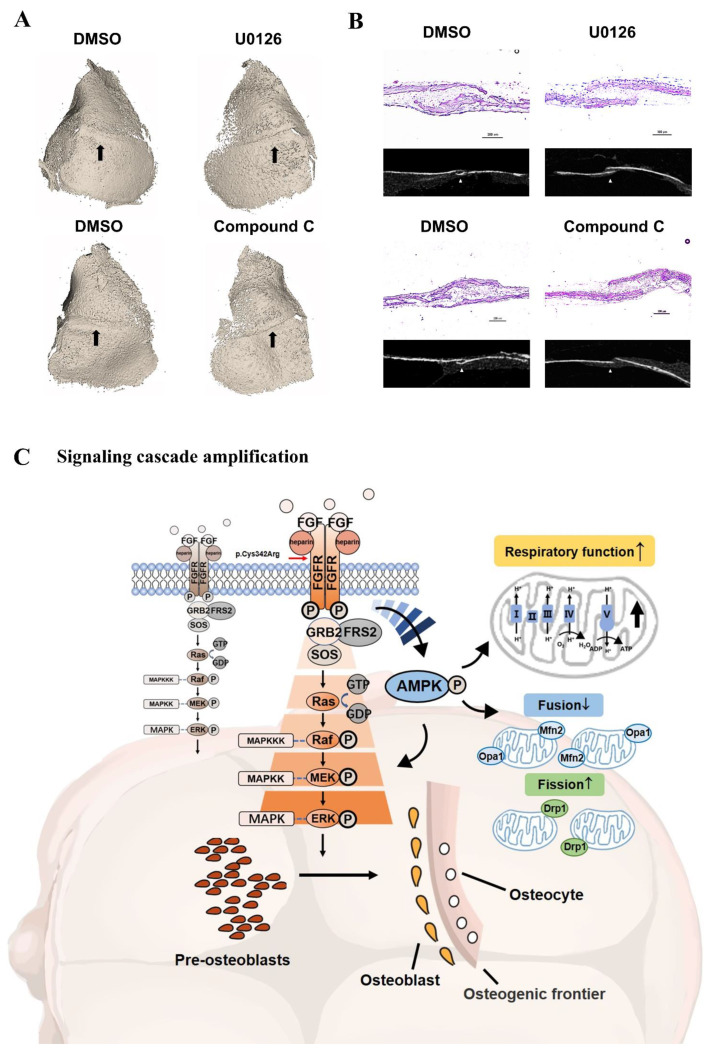
(**A**) U0126 and Compound C attenuated the closure of coronal sutures of cultured calvarias. It was shown by 3D reconstruction images that there was an obvious linear gap between the frontal and parietal bone after the treatment with U0126 and Compound C. In the control groups, dense bony unions were formed in coronal suture regions. The coronal suture region was marked by a black arrow. (**B**) As shown by the section selected at the corresponding position of micro-CT, there were gaps between the osteogenic fronts of frontal and parietal bone in the inhibitor groups. The osseous cross-linking in the coronal suture was closer in control groups. As revealed by H&E staining, the synostosis between frontal and parietal bone presented as serrated in the control. However, there was a decreased overlapping region after treatment with U0126 and Compound C. (**C**) Schematic diagram of osteogenesis and internal mechanism. In the process of pre-osteoblast cells further differentiating, the relatively complete FGF/FGFR2-AMPK-Erk1/2 MAPK pathway plays a pivotal role. The constitutive activation of the AMPK-Erk1/2 signaling path network leads to enhanced osteogenesis because of the FGFR2 mutation (p.Cys342Arg) differing from wild-type FGFR2. In addition to being upstream of the Erk1/2 signal, the expression and activation of AMPK amplifies the mitochondria respiratory function and changes the ratio of mitochondrial division and fusion to respond to osteogenic activity. The differentiated osteoblasts then secrete the extracellular matrix and subsequently promote mineral deposition at the front edge of osteogenesis, which may lead to the malformation of cranial vault bones in Crouzon syndrome.

## Data Availability

The data are contained in the article.

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
