# Peer review of "FGFR2 Mutation p.Cys342Arg Enhances Mitochondrial Metabolism-Mediated Osteogenesis via FGF/FGFR-AMPK-Erk1/2 Axis in Crouzon Syndrome"

_cells, 2022, doi:10.3390/cells11193129_

Round 1

Reviewer 1 Report

The manuscript by Wang et al reports the original description of a detailed in silico, in vitro, ex vivo and in vivo characterization of the FGFR2 c.T1024C mutation in the FGFR2 gene causing the p.C342R substitution, which is known to be a pathogenic variant causing Crouzon syndrome.

The study is very well conducted and the experiments performed enable proving the involvement of mitochondrial dysfunctions caused by the altered and costitutively active downstream signaling associated with the mutated receptor.

I have only a few minor comments and suggestions to propose to the

Authors, to improve the clarity and overal quality of the manuscript, and to deem it suitable for publication:

- Though no previous detailed functional characterization of the mutation pathogenicity has been performed, to the best of my knowledge, the Authors should clarify that the p.C342R mutation in the FGFR2 gene is well known as a causative mutation for Crouzon syndrome, being annotated as pathogenic in ClinVar since 2020 (https://www.ncbi.nlm.nih.gov/clinvar/RCV000014177/). I think that this clarification deserves a few lines of discussion with appropriate review of the literature describing the same mutation in previous cohorts (see PMID: 7987400,7719345,8528214,10633130).

- Please use the correct HUGO coding conventions, according to which gene symbols need to be in italics and only proteins do not require italics.

- Check for incoherence and review this sentence in the introduction “Fibroblast growth factor receptor 2 (FGFR2) or 4 FGFR3 mutations are considered as the genetic etiology of Crouzon syndrome. Unlike 5 FGFR2, FGFR3 mutation is thought to be related to Crouzon syndrome with acanthosis 6 nigricans”

- some minor english style revision could be implemented to improve the readibility of the manuscript.

Reviewer 2 Report

In this work Wang and co. explore the effect of a recurrent mutation in FGFR2 on osteogenesis on an heterologous murine model. While the work is interesting and approaches the problem from multiple angles, including cell growth, differentiation, and mitochondrial function, the work as it is has some major flaws. While some of the results are interesting, they are poorly supported and interpretation have to be more cautious. All the article needs a strong tone-down and some major issues should be addressed.

The authors seem to omit most of the relevant previous work on the area, including other functional studies on the very same variant (PMID: 27596331). Also, Cys342 is one of the most commonly mutated residues with at least 7 different variants and more than 20 publications, some of them also performing functional analysis). The authors do not mention any of them, not even that the change has been previously identified.  The authors do not mention number of replicates in any of the experiments and most of the results relay on poor quality WB that are not quantified (if the number of replicates allow that) and analyzed statistically, so their conclusions are not sustained by the presented data. CCK8 experiment on each section show reduced proliferation, not inhibited as the authors say.  

In addition, to this reviewer, the correct expression of the expression vectors has not been properly demonstrated (see comments on point 3.5) and this is a main issue as the rest of experiments relay on being confident that the heterologous proteins are working appropriately and that the tag is not affecting folding and/or intracellular localization of the protein. I strongly suggest improving the tag detection and assessing the protein localization by immunocytochemistry.  Also, appropriate controls are not always included, for example in the DN experiments the WT control is omitted, and the logical comparison with the siRNA is not performed. So, the authors can’t conclude that the mutation leads to an increase of FGFR2 activity.

Moreover, gain of function will mean that a new function is performed by the protein, I think the authors mean hyperactivity or hyperactivation.

Mutation nomenclature should follow HGVS rules, p.C342R should be p.Cys342Arg and so on along the whole text. Variant c.T1024C should read c.1024T>C…

MC3T3-E1 is a murine model and should be stated very clearly, even in the abstract, for example, in line 28 (page 1) it can be stated as: “were constructed and transfected into pre-osteoblastic MURINE cell line MC3T3-E1 cells”.

Introduction:

As the article is going to focus mainly on FGFR2 role in osteogenesis and its regulated pathways further revision on previously published work could be useful in this section.

Material and methods:

This section needs improvement, with number of replicates, full names of genes (such as COL1A2 not COL1) and a section on the obtention of the expression plasmids, especially the domain negative vector. Is the tag N or C terminal?

Results Section:

Appendix figure A can be omitted.

Point 3.2:

Please, indicate genomic position of the variant, transcript of reference, absence in databases… The fact that the variant in in ClinVar and has been extensively reported in the literature must be mentioned.

Point 3.3: the wording of this section is specially confusing and needs rewriting for clarification. Is MT mutant? Please, clarify.

The second paragraph of point 3.3 states that there is no increase in t-FGFR2. I understand that it is the result on MC3T3 cell lines after transfection of the wt and mutant vectors but not transfected or empty vector transfected cell lines has not been included and it is an important omission.  This should be addressed properly if possible. If these experiments are impossible to perform, this limitation must be mentioned and properly discussed.  

Point 3.4:

Regarding “CCK-8 showed that knockdown of FGFR2 suppressed the proliferative activity of MC3T3-E1 cell lines (Fig. 3B)”. In the figure a moderate reduction is shown, NOT a suppression.

Figure 3: What does NC means?

Point 3.5: the plasmid construction, specially the DN construction is totally unclear. Appendix figure 3 is too small to see the representation and it have to be clarified exactly the deletion performed (from X to Y residues). The WB presented in App.Figure 3 are blurry, small and do not demonstrate a correct expression of a unique band. To me, it is unacceptable that a Flag-tag result is so unclear and the WB images have been drastically cropped, even when no lack of space justify the cropping. Also, there is no weight marker or load control.  The correct expression of the FGFR2 isoforms is not demonstrated to this reviewer so this point could not be evaluated.

In addition, the fact that the wt is not included in these experiments is totally misleading and do not allow correct interpretation of the results. Are the DN results similar to the siRNAFGFR2 or like the WT?

Point 3.6:  Page 13, line 18: U0126 is NOT specific (see PMID: 36091807 for a review of other pathways affected).

Figure 4: I think there is a mistake in the legend and panels B and C are mixed. There is also a type in panel C (compund)

Discussion:

The discussion need to address previous data on FGFR2 functional studies, as well as to review previous mentions to the p.Cys342Arg variant and other similar variants functionally analyzed by other authors, as previously mentioned, and to tone it down according to the presented data. FGFR2 pathways are extremely complex and this work address only few of them, caution is needed.  

Round 2

Reviewer 2 Report

The WB can be improved but is ok to me.